# The deubiquitinase MYSM1 dampens NOD2-mediated inflammation and tissue damage by inactivating the RIP2 complex

Swarupa Panda[1] & Nelson O. Gekara[1]

NOD2 is essential for antimicrobial innate immunity and tissue homeostasis, but require tight regulation to avert pathology. A focal point of NOD2 signaling is RIP2, which upon poly-ubiquitination nucleates the NOD2:RIP2 complex, enabling signaling events leading to inflammation, yet the precise nature and the regulation of the polyubiquitins coordinating this process remain unclear. Here we show that NOD2 signaling involves conjugation of RIP2 with lysine 63 (K63), K48 and M1 polyubiquitin chains, as well as with non-canonical K27 chains. In addition, we identify MYSM1 as a proximal deubiquitinase that attenuates NOD2:RIP2 complex assembly by selectively removing the K63, K27 and M1 chains, but sparing the K48 chains. Consequently, MYSM1 deficient mice have unrestrained NOD2-mediated peritonitis, systemic inflammation and liver injury. This study provides a complete overview of the polyubiquitins in NOD2:RIP2 signaling and reveal MYSM1 as a central negative regulator restricting these polyubiquitins to prevent excessive inflammation.

[1] The Laboratory for Molecular Infection Medicine Sweden (MIMS), Umeå Centre for Microbial Research (UCMR), Umeå University, 90 187 Umeå, Sweden. Correspondence and requests for materials should be addressed to N.O.G. (email: nelson.gekara@mims.umu.se)

NOD2 is an intracellular innate immune receptor that detects a broad array of danger signals, including bacteria, viruses, and parasites, as well as distressed intracellular organelles[1–4]. NOD2 triggering activates the nuclear factor-kappaB (NF-kB) and mitogen-activated protein kinase (MAPK) pathways to induce the expression of inflammatory cytokines and antimicrobial peptides, which coordinate pathogen clearance and subsequent restoration of tissue homeostasis. However, dysregulation in NOD2 signaling can cause immunopathology and may engender a proneness to infections, inflammatory diseases, and cancer[1,4,5]. The host regulatory mechanisms that ensure an optimal activation of NOD2-mediated inflammation are not fully understood.

Ubiquitination is a reversible post-translational modification that controls nearly all cellular processes by regulating the activity, localization, and half-life of proteins[6,7]. It involves the attachment of ubiquitin (Ub) onto lysine residues on target protein substrates at one or multiple sites. Moreover, ubiquitins can be linked to each other via either the N-terminal methionine (M1), or the seven internal lysine residues (K6, K11, K27, K29, K33, K48, K63), giving rise to eight types of polyubiquitin chains with different conformations and functions. The best-characterized polyubiquitin chains include K48-linked chains that target proteins for proteasomal degradation[8] and M1- or K63-linked chains that mediate protein–protein interactions during signal complex assembly[9–14]. The current knowledge of the ubiquitin system mostly relates to homotypic polyubiquitins (contain single linkage type). However, on top of this complexity, ubiquitins can be attached to each other via two or more lysine residues to form heterotypic polyubiquitin chains with multiple linkages that adopt mixed or branched topology[7].

Upon activation, NOD2 oligomerizes and binds the proximal adaptor receptor-interacting protein kinase 2 (RIP2/RIPK2/RICK)[15]. This results in the recruitment of E3 ligases XIAP[16,17], ITCH[18], and Pellino3[19] which subsequently conjugate K63-polyubiquitin chains onto RIP2[20]. These polyubiquitin chains then provide docking sites for other signaling proteins. This includes, for instance, the linear ubiquitin chain assembly complex (linear ubiquitin chain assembly complex (LUBAC); a tri-meric complex composed of SHARPIN, HOIL-IL, and the catalytic subunit HOIP) that conjugates M1-polyubiquitin chains to RIP2[16]. The K63- and M1- polyubiquitinated RIP2 provides a platform to which TAK1, TAB2/3, and the IKK kinase complex (IKKγ (NEMO) and IKKα/β)[21,22] are recruited, resulting in their activation to mediate downstream NF-kB- and MAPK-driven inflammatory responses.

In spite of this progress, our understanding of the ubiquitin modifications involved in NOD2 signaling and how they are regulated to avert pathology is still incomplete. For instance, aside from K63- and M1 ubiquitin chains, does the NOD2: RIP2 signaling complex involve other polyubiquitin chains? And if so, what is their architecture? Moreover, how are the proximal ubiquitination events in NOD2:RIP2 complex counteracted to ensure a timely resolution of NOD2 signaling?

Here we show that activation of NOD2 involves the attachment of K63, K48, and M1 polyubiquitin chains to RIP2, as well as attachment of non-canonical K27 chains, heretofore not associated with NOD signaling. Our data further demonstrate that the three non-degradative (K63, K27, and M1) chains are part of heterotypic ubiquitin oligomers covalently attached to RIP2 via K209. Finally, we identify the H2A deubiquitinase myb-like SWIRM and MPN domains 1 (MYSM1, also known as 2A-DUB or KIAA1915) as an upstream deubiquitinase (DUB) that targets all three non-degradative ubiquitin chains to disrupt the NOD2: RIP2 complex. Importantly, in mice, MYSM1 deficiency causes hyper-inflammation characterized by enhanced cytokine production, mobilization of inflammatory cells into tissues and enhanced susceptibility to NOD2-mediated tissue injury.

## Results

### NOD2 evokes K63, K48, M1, and K27 polyubiquitination of RIP2.
To comprehensively elucidate the types of polyubiquitin chains attached to RIP2 and their architecture, ubiquitinated proteins were purified from bone marrow derived macrophages (BMDMs) stimulated with the lipidated NOD2 ligand mur-amyldipeptide (L18-MDP) by TUBE (tandem ubiquitin-binding entity) pulldowns, then subjected to ubiquitin restriction (Ubi-CRest) analysis[23]. These digestions were then analyzed by immunoblotting for RIP2 to reveal the sensitivity of ubiquitinated RIP2 (Ub-RIP2) to different linkage specific DUBs (Fig. 1a, b). Incubation with the non-specific DUB USP2 resulted in complete stripping of ubiquitins from Ub-RIP2, causing a dramatic drop from the high MW Ub-RIP2 smear to the non-ubiquitinated RIP2 (Fig. 1a compare lane 3 to lane 4). In contrast, Ub-RIP2 was completely insensitive to OTUD3 (K6-, K11-specific) and Cez-zane (K11-specific) (Fig. 1a compare lane 3 to lanes 5 and 6). Notably, incubation with OTULIN (M1-specific), AMSH (K63-specific), OTUB1 (K48-specific), UCHL3 (K27-specific) or TRABID (K29-, K33-, and K63-specific), resulted in a substantial, but incomplete, drop from high to lower MW Ub-RIP2 forms (Fig. 1a compare lane 3 to 7–11). In this regard, it is worth to point out that among the limitations of the UbiCrest include enzyme specificities, lack of enzymes that only cleave K33 or K29 and reaction conditions. These caveats notwithstanding, and whereas future validation by mass spectrometry should be carried out, the above data indicate that in response to NOD2 triggering, RIP2 undergoes K63-, K27-, M1-, and K48-, but not K6-, K11- and most likely not K29- or K33-linked polyubiquitination.

To elucidate the architecture of identified polyubiquitin chains, we probed the UbiCRest digestions with specific antibodies against the identified ubiquitin linkages (Fig. 1c). We found that cleavage of K63-linked chains (by AMSH or TRABID) also caused a drop in M1- and K27- but not K48-linked chains (Fig. 1c compare lane 3 to lanes 8, 10). First, these results show that K63- and K48-linked chains are attached to RIP2 via different lysine residues. Secondly, they demonstrate that K63-polyubiquitins are the proximal chains onto which M1 and K27 chains are conjugated. Consistent with this, UCHL3-mediated digestion of K27 chains had no effect on K63 or K48 chains, but resulted in an appreciable drop in M1 (Fig. 1c compare lane 3 to lane 7). On the other hand, digestion with OTULIN only resulted in a drop in the M1-chains without affecting K63, K27, and K48 chains (Fig. 1c compare lane 3 to lane 11), indicating that M1 chains are attached both to the proximal K63 chains and its K27 branches.

Next we wanted to understand the sequence by which K63, K48, K27, and M1 chains are attached to RIP2 and their contribution to RIP2-dependent signaling. Towards that end, we blocked LUBAC-mediated M1 ubiquitination using gliotoxin (GT)[24] and performed RIP2 pulldowns to assess complex formation as well as activation of downstream signaling events. As expected, GT completely blocked MDP-induced attachment of M1 polyubiquitins to RIP2, as well as activation of IKKα/β and cytokine expression (Fig. 1d and Supplementary Figure 1). However inhibition of M1 polyubiquitination did not affect the conjugation of K63-, K27-, or K48 chains to RIP2 or the concomitant phosphorylation of RIP2 and TAK1 (Fig. 1d and Supplementary Figure 1). Together, as schematically summarized in Fig. 1e, these data demonstrate that NOD2 signaling involves K63-, K27-, M1-, and K48– linked polyubiquitination of RIP2. Secondly, they show that K63-, K27-, and M1-linked

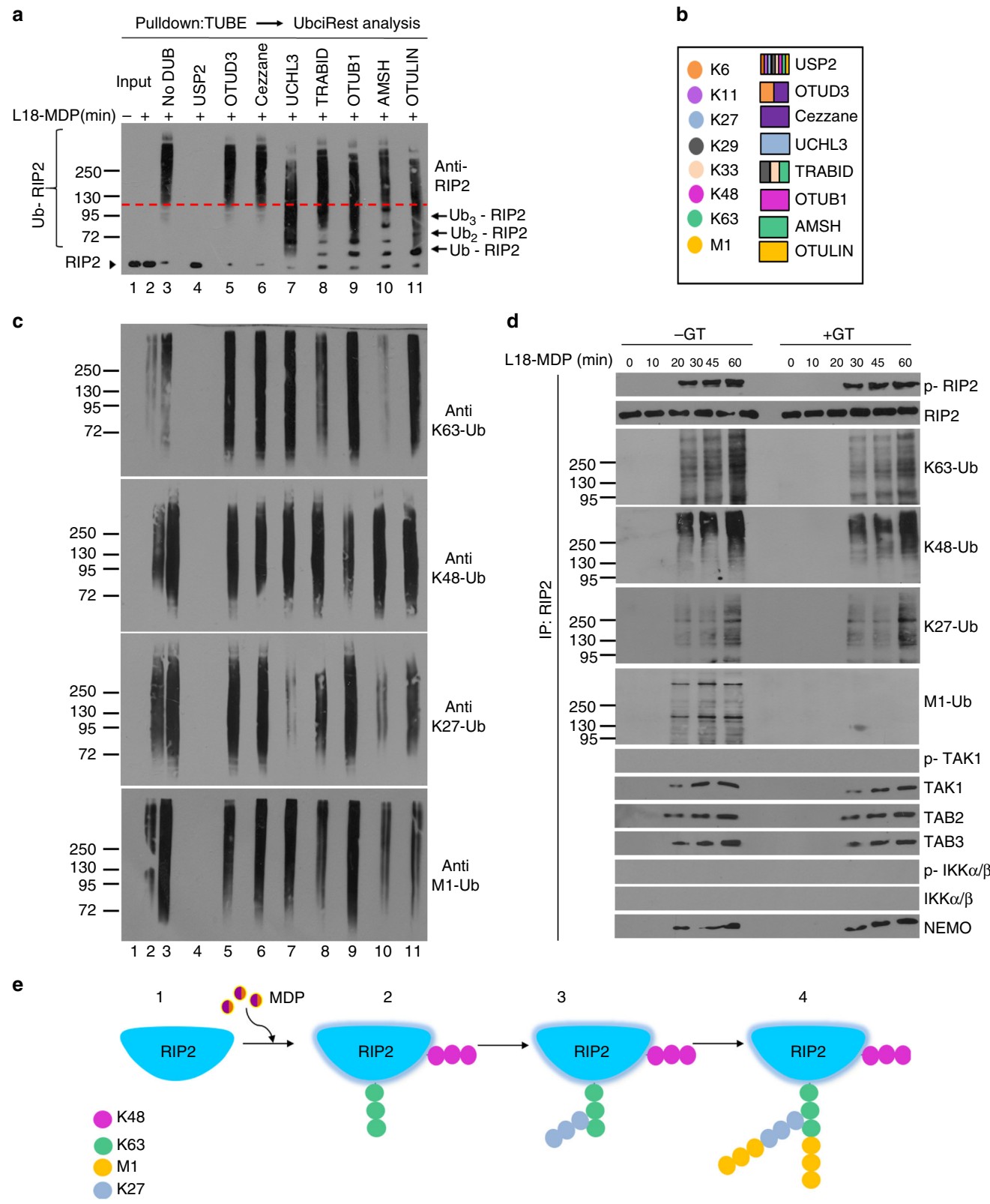

polyubiquitin chains are part of a hybrid polymers with K63 chains being the proximal chains onto which K27 and M1 chains are attached in that order (Fig. 1e). Moreover, our data also show that K48-Ub chains are attached onto RIP2 via a different lysine residue than the K63-associated hybrid Ub-polymers. Finally, they also show that, although essential for the downstream

NOD2-mediated activation of NF-kB, LUBAC-mediated conjugation of M1 chains is dispensable for the proximal polyubiquitination events that coordinate the assembly of the NOD2:RIP2 complex (Fig. 1d and Supplementary Figure 1).

The lysine residue K209 in RIP2 has been proposed to be a major site of polyubiquitination implicated in regulation of

**Fig. 1** NOD2 signaling involves K63-, K48-, K27- and M1- polyubiquitination of RIP2. **a** Determination of polyubiquitin linkages on RIP2 by ubiquitin restriction (UbiCRest) analysis. Ubiquitinated proteins isolated by TUBE pulldowns from WT BMDMs stimulated with L18-MDP for 1 h were incubated with indicated DUBs for 1 h. Samples were then immunoblotted for RIP2. The red dashed line is an arbitrary line highlighting a relative shift from high to lower MW intermediate /complete digestion products of Ub-RIP2. **b** Color codes of ubiquitin linkage specificity of DUBs used in the UbiCRest analysis in **a, c**. **c** Corresponding UbiCRest digestions depicted in **a**, analysed by immunoblotting for K63, K48, K27, and M1 linkages. **d** Cell lysates from WT BMDMs stimulated with L18-MDP for indicated duration in the presence (+) or absence (−) of the LUBAC inhibitor gliotoxin (GT) were immunoprecipitated with anti-RIP2. Pulldowns were then immunoblotted for indicated molecules. **e** Schematic summary of the proposed sequence by which different polyubiquitins are conjugated to RIP2. Data are representative of at least 3 independent experiments. See also Supplementary Figure 1 and Supplementary Figure 2

NOD2 signaling[18,20]. To elucidate the role of the K209 residue in NOD2 signaling, we analyzed cytokine response and polyubiquitination events in $Rip2^{-/-}$ BMDMs complemented with full-length RIP2 or the RIP2 (K209R) mutant where K209 was substituted with arginine. BMDMs expressing RIP2(K209R) were completely unresponsive to L18-MDP-induced polyubiquitination and cytokine (TNFα) response (Supplementary Figure 2a-c). These results confirm K209 as the proximal lysine residue on RIP2 onto which the K63, K27, and M1 chains are attached. Further, they demonstrate that K209- attached chains are essential for the subsequent conjugation of K48 chains—likely by providing docking sites for signaling components that mediate the K48-linked polyubiquitination of RIP2 (Supplementary Figure 2d).

**MYSM1 is recruited to NOD2:RIP2 independently of LUBAC**. Thus far the negative regulation of NOD2 signaling has been ascribed to three major components: the deubiquitinases (DUBs) CYLD[25–28], OTULIN[29–32], and the ubiquitin-editing enzyme A20[25,33]. However the effect of these regulators on the NOD2: RIP2 complex is indirect and depends on the linear ubiquitin chain assembly complex linear ubiquitin chain assembly complex (LUBAC) for them to interact with and disrupt the NOD2:RIP2 complex[25–28,31,33]. Therefore, a key conceptual gap in our knowledge of the NOD2 pathway is whether and how proximal signaling events coordinating the NOD2:RIP2 complex assembly are also controlled prior to LUBAC recruitment.

The deubiquitinase MYSM1 is a key component of the epigenetic signaling machinery[34,35], but accumulates in the cytoplasm in response to infection or TLR signaling[36]. But, whether MYSM1 is involved in NOD2 signaling is not known. While exclusively in the nucleus at steady state, we found that in response to L18-MDP, MYSM1 rapidly accumulated in the cytoplasm of BMDMs and interacted with RIP2 (Fig. 2a, b). Interestingly, inhibition of LUBAC activity by GT neither affected the recruitment of MYSM1, HOIP (a component of LUBAC), and CYLD to the RIP2 complex, nor the phosphorylation of RIP2, TAK1, and p38 MAPK. However, as expected, GT abolished the recruitment of the M1 chain-stabilizer A20[25,37] (Fig. 2b). Upon immunoprecipitation, we observed no direct interaction between MYSM1 with HOIP, NEMO or the negative regulators CYLD, OTULIN or A20 (Fig. 2c). This indicates that MYSM1 is part of the NOD2:RIP2 complex, but that unlike A20, MYSM1 does not require LUBAC activity for its recruitment to this complex.

**MYSM1 attenuates the NOD2:RIP2 complex assembly**. The above observations prompted us to investigate whether MYSM1 might be involved in the regulation of NOD2 signaling. In response to both the NOD2 ligand L18-MDP or the NOD1 ligand γ-d-glutamyl-meso-diaminopimelic acid (iE-DAP), BMDMs from $Mysm1^{-/-}$ mice elicited a more robust cytokine (TNFα and IL6) response, while, as expected, $Rip2^{-/-}$ BMDMs remained

unresponsive (Fig. 3a–d). Moreover, in response to L18-MDP, $Mysm1^{-/-}$ BMDMs exhibited enhanced phosphorylation of RIP2, TAK1, IKK-α/β, and p38 MAPK compared to the WT or the unresponsive $Rip2^{-/-}$ BMDMs. This demonstrated that MYSM1 was impeding proximal NOD2 signaling events (Fig. 3e).

Next we assessed the impact of MYSM1 on the NOD2:RIP2 complex assembly. We found that in the absence of MYSM1, the recruitment of proximal NOD2 signaling components XIAP, HOIP, TAK1, TAB2, TAB3, and NEMO to the NOD2:RIP2 complex and the concomitant phosphorylation of RIP2, TAK1, IKKα/β was enhanced (Fig. 4a, c), indicating that MYSM1 was acting upstream of these molecules to suppress NOD2 signaling. To explore further, we assessed by immunoprecipitation the interaction of MYSM1 with the other components in the NOD2: RIP2 complex. Although MYSM1 interacted with NOD2, RIP2, and XIAP, consistent with the above data (Fig. 2c), we observed no direct interaction between MYSM1 and HOIP, TAK1, TAB2, TAB3, or the IKK complex (Fig. 4b). These data indicated that MYSM1-mediated negative regulation of NOD2 signaling was likely via an upstream effect on NOD2:RIP2 complex formation rather than via an effect on the downstream components recruited into this complex (Fig. 4d).

Previously we reported that MYSM1 can deubiquitinate TRAF6 and TRAF3—central nodes in the Toll-like receptor (TLR) signaling[36]. Thus we sought to determine whether MYSM1-mediated suppression of NOD2 signaling was via TRAF6. To do that, we employed SMAC mimetic LCL-161. Although causing depletion of cIAP1/2 proteins[38] and blocking the recruitment of TRAF6 into the NOD2:RIP2 complex, in agreement with previous observations[39], LCL-161 had no substantial effect on NOD2 signaling (Supplementary Figure 3). Importantly, LCL-161 did not abolish the enhanced MDP-induced NOD2:RIP2 complex assembly or phosphorylation of RIP2 and IKKα/β caused by MYSM1 deficiency (Supplementary Figure 3). Thus, whereas initial overexpression studies had previously suggested a possible role of TRAF6 in NOD2 signaling[40], it appears that TRAF6 is non-essential for NOD2 signaling, a view consistent with genetic studies showing that TRAF6 deficient cells are not defective in NOD1/ NOD2 signaling[20,41,42]. In brief, these results indicate that MYSM1-mediated suppression of the NOD2 pathway is not mediated via TRAF6 or its signaling partners cIAP1 and cIAP2, but most likely through the direct inactivation of the NOD2:RIP2 complex.

To elucidate in more details how MYSM1 interacts with and modulates the NOD2:RIP2 complex, next we asked whether ubiquitin chains on RIP2 were required for the recruitment of MYSM1. Although both MDP and the TLR4 ligand lipopolysaccharide (LPS) can induce cytoplasmic accumulation of MYSM1, LPS stimulation does not result in RIP2 ubiquitination (Supplementary Figure 4a–c). Interestingly, in contrast to MYSM1 in MDP-stimulated cells, cytoplasmic MYSM1 in LPS-stimulated cells showed no interaction with RIP2 (Supplementary Figure 4a–c). These data demonstrate that recruitment of

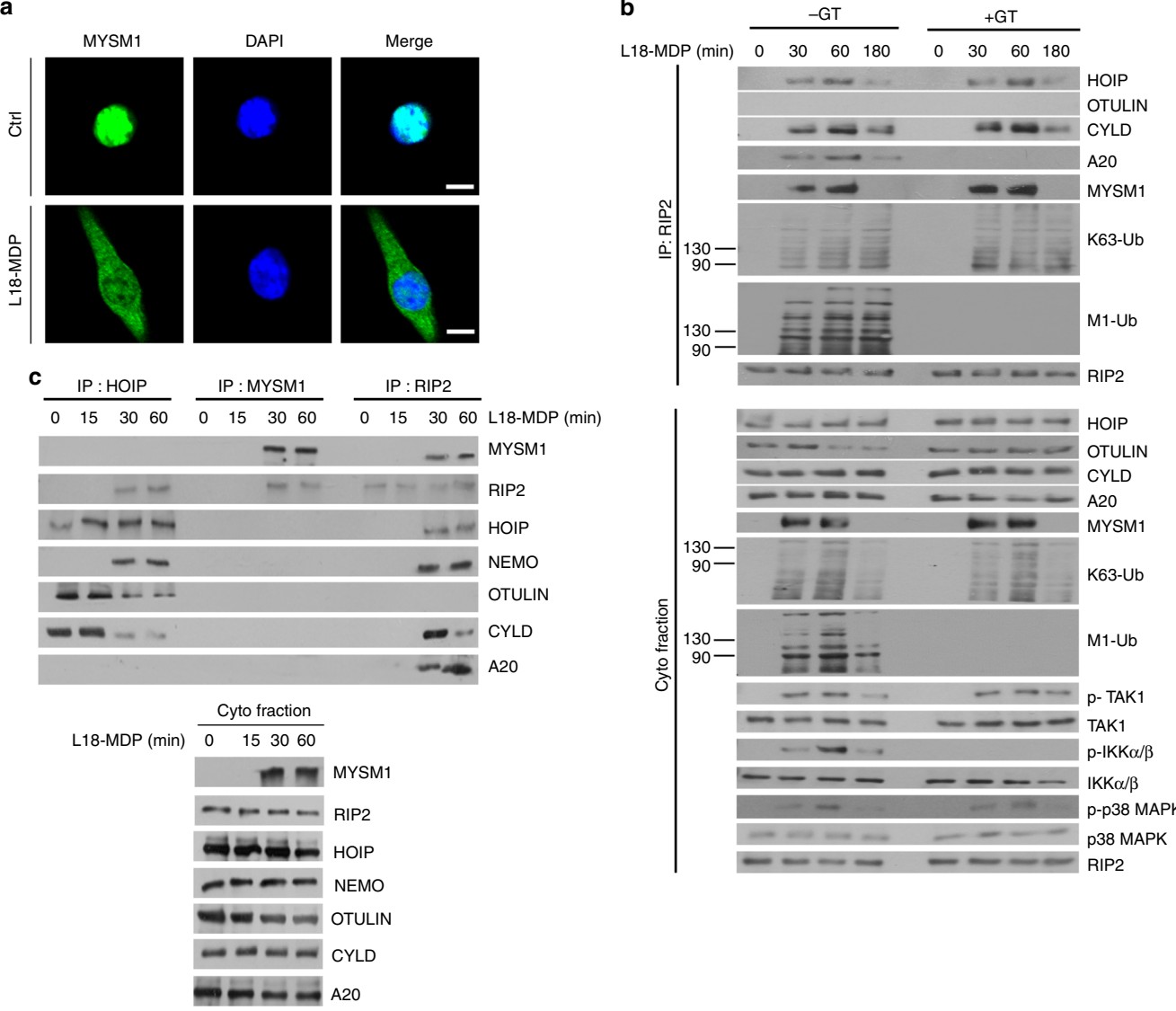

**Fig. 2** MYSM1 is recruited to the NOD2:RIP2 complex independently of LUBAC. **a** Immunofluorescence staining of MYSM1 (green) in BMDMs before (Ctrl) or after 1 h stimulation with L18-MDP. Scale bar (10 μm). **b** RIP2 immunoprecipitates and corresponding whole cytoplasmic fractions from WT BMDMs stimulated with L18-MDP for indicated durations in the presence or absence of gliotoxin (GT) were immunoblotted for indicated molecules. **c** HOIP, MYSM1, or RIP2 immunoprecipitates from WT BMDMs stimulated with L18-MDP for different durations, as well as corresponding whole cytoplasmic fractions, were examined for the presence of indicated molecules. Data are representative of three independent experiments

MYSM1 to the NOD2:RIP2 complex is controlled by at least two layers of regulation: 1) an initial inflammatory trigger to induce cytoplasmic accumulation of MYSM1 and 2) the presence of ubiquitin chains on RIP2.

**MYSM1 removes K63, K27, and M1 polyubiquitins from RIP2.** Next we investigated whether disruption of the NOD2:RIP2 complex by MYSM1 was via the removal of ubiquitin chains from RIP2. To that end, first we isolated total-Ub, K63-linked Ub and M1-Ub conjugates from L18-MDP-stimulated WT and *Mysm1*$^{-/-}$ BMDMs using TUBE, K63-TUBE, or M1-TUBE, respectively. Gel separation and immunoblot analysis of these isolates revealed that Ub-RIP2 from *Mysm1*$^{-/-}$ were more abundant and migrated at higher MW (Supplementary Figure 5a) than those isolated from WT BMDMs, indicating that MYSM1 is an antagonist of RIP2 ubiquitination.

Next we purified ubiquitin conjugates from L18-MDP-stimulated *Mysm1*$^{-/-}$ BMDMs by TUBE and incubated them with recombinant MYSM1 (Rec-MYSM1) and compared the ubiquitin restriction pattern of Ub-RIP2 with that of AMSH, UCHL3, OTULIN or a combination of these three DUBs (Fig. 5a, b). The restriction pattern of MYSM1 was comparable to that by AMSH or the combination of AMSH+UCHL3+OTULIN (Fig. 5b, compare lane 3 to lanes 5, 8, 9) demonstrating that MYSM1 was either cleaving the proximal K63 chains on RIP2 or that it was additionally trimming the attached K27 and M1 branches.

To further interrogate these findings, we immunoprecipitated RIP2 from native or denatured (boiled in SDS to disrupt noncovalent interactions) cytoplasmic fractions of L-18-MDP-stimulated WT and *Mysm1*$^{-/-}$ BMDMs and probed them with antibodies against the specific ubiquitin linkages. We found that in the absence of MYSM1, K63-, K27-, and M1-linked

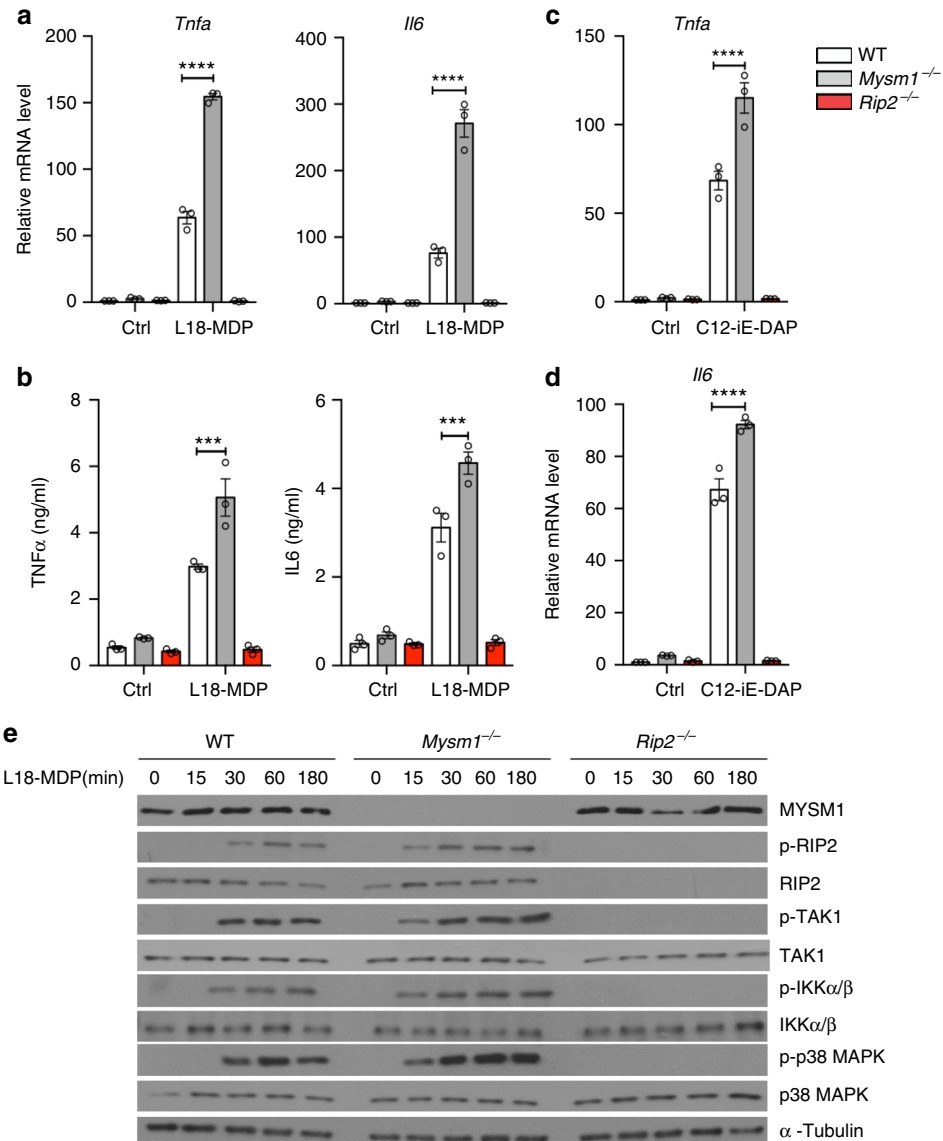

**Fig. 3** MYSM1 suppresses proximal NOD2 signaling events and cytokine response. **a**, **b** WT, *Mysm1*$^{-/-}$, and *Rip2*$^{-/-}$ BMDMs, either unstimulated (Ctrl) or stimulated with L18-MDP (200 ng/ml), were analyzed by qRT-PCR for *Tnfa* and *Il6* transcript 6 h post stimulation (**a**) or for secreted TNF-α and IL-6 12 h post stimulation (**b**). **c**, **d** WT, *Mysm1*$^{-/-}$, and *Rip2*$^{-/-}$ BMDMs, either unstimulated (Ctrl) or stimulated with C12-iE DAP (200 ng/ml), were analyzed by qRT-PCR for *Tnfa* (**c**) and *Il6* (**d**) transcript 6 h post stimulation. Results in (**a**–**d**) are from three independent experiments. Data are shown as mean ± s.e.m. ($n = 3$). ***$p < 0.001$ determined by one-way ANOVA followed by Bonferroni's post-test depicts statistical significance. **e** WT, *Mysm1*$^{-/-}$ and *Rip2*$^{-/-}$ BMDMs, either unstimulated (Ctrl) or stimulated with L18-MDP for indicated duration, were analysed by immunoblotting for MYSM1, p-RIP2, RIP2, p-TAK1, TAK1, p-IKKα/β, IKKα/β, p-p38 MAPK, and p38 MAPK. α- tubulin was used as loading control. Data in (**e**) are representative of three independent experiments

ubiquitination of RIP2 was enhanced while K48-linked ubiquitination was unaffected (Fig. 5c). In further support of these findings, incubation of Ub-RIP2 immunoprecipitated from *Mysm1*$^{-/-}$ BMDMs with Rec-MYSM1 resulted in diminished K63-, K27-, and M1-linked, but not K48-linked, ubiquitin chains (Supplementary Fig. 5b). In contrast, Rec-MYSM1 had no effect on ubiquitin chains attached to NEMO (Supplementary Fig. 5c), indicating that MYSM1 was selectively cleaving K63, K27, and M1, but not K48 linkages, from RIP2.

To gain a complete overview of the ubiquitin linkage specificity of MYSM1, we incubated Rec-MYSM1 with Di-ubiquitins (Ub$_2$) of different linkages. MYSM1 was found to cleave M1, K6, K27, and K63 linkages, but not the K11, K29, K33, or K48 linkages (Supplementary Figure 5e and Fig. 5d).

The above data show that NOD2 signaling involves the attachment of K63, K27, M1, and K48 chains to RIP2 and that MYSM1 interacts with and removes K63, K27, and M1 but not K48 chains from RIP2. Therefore, to independently verify these findings, we asked whether incubation of Ub-RIP2 isolated from MDP-stimulated cells with a mixture of MYSM1+OTUB1 could get rid of all ubiquitins from RIP2. Indeed MYSM1+OTUB1 completely stripped all ubiquitins from RIP2 (Fig. 5e). Together, these data conclusively demonstrate that K63-, K27-, M1-, and K48-linked polyubiquitins are the only chains conjugated to RIP2 upon NOD2 activation and that MYSM1-mediated disruption of the NOD2:RIP2 complex is due to its ability to selectively remove the K63, K27, and M1 chains, but not the degradative K48 chains, from RIP2 (Fig. 5f).

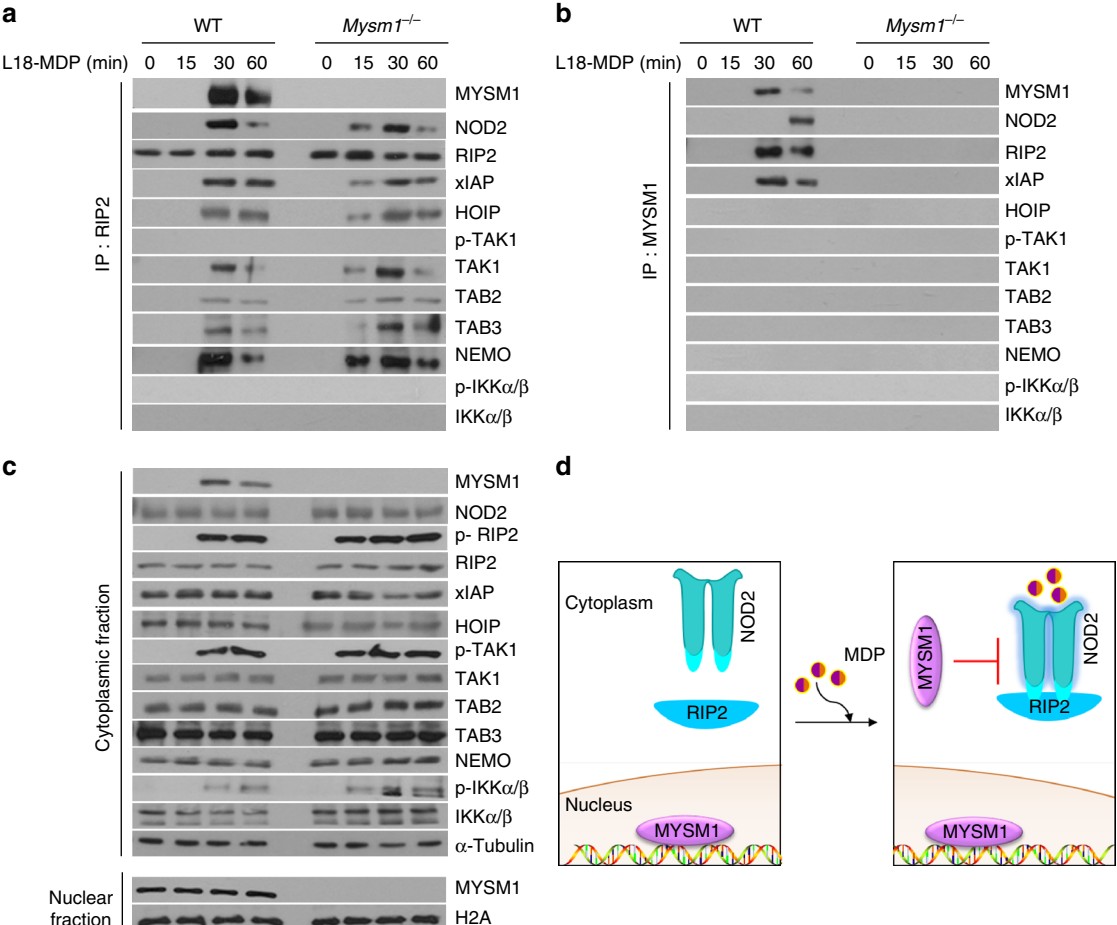

**Fig. 4** MYSM1 interacts with and disrupts NOD2:RIP2 complex. **a** Cytoplasmic fractions from WT and *Mysm1*−/− BMDMs stimulated with L18-MDP for indicated duration were immunoprecipitated with anti-RIP2 (**a**) or anti-MYSM1 (**b**). The pulldowns (**a, b**) and corresponding cytoplasmic or nuclear fractions (**c**) were immunoblotted for indicated proteins. **d** Schematic representation of MYSM1 subcellular localization before and after MDP stimulation. Data are representative of at least three independent experiments

**MYSM1 attenuates NOD2 signaling via SWIRM and MPN domains**. MYSM1 contains a central SWIRM (SWI3, RSC8, and MOIRA) domain and a C-terminal MPN (JAB/Mov34) domain that contains the deubiquitinase activity[35,36] (Fig. 6a). To define the MYSM1 motifs involved in the negative regulation of NOD2 signaling, we transduced *Mysm1*−/− BMDMs with His-tagged full-length MYSM1 (MYSM1-FL), or MYSM1 mutants lacking either the SWIRM domain (ΔSWIRM, aa 363–464) or the MPN domain (ΔMPN aa 563–673), or a fusion construct of SWIRM and MPN (SWIRM+MPN). Much like the endogenous form, MYSM1-FL, ΔSWIRM, and ΔMPN were localized to the nucleus at steady-state conditions, but accumulated in the cytoplasm upon MDP stimulation (Fig. 6b, d and Supplementary Figure 6a). This was in contrast to the SWIRM+MPN that was exclusively cytoplasmic (Fig. 6b, d and Supplementary Figure 6a). Complementation of *Mysm1*−/− BMDMs with MYSM1-FL or SWIRM+MPN suppressed L18-MDP-induced recruitment of TAK1, TAB2, TAB3, and NEMO to the RIP2 complex (Fig. 6c and Supplementary Figure 6a), thereby inhibiting the activation of TAK1, IKKα/β, and p38 MAPK (Fig. 6d) and the concomitant cytokine induction (Fig. 6e, f Supplementary Figure 6b). In contrast, ΔSWIRM or ΔMPN did not exhibit such inhibitory effect (Fig. 6c-f and Supplementary Figure 6a-c), demonstrating that MYSM1 interacts with and inactivates the RIP2 complex via the SWIRM and MPN domains.

To explore whether enzymatic activity within the MPN domain was required for the interaction with RIP2, we mutated the aspartate residue at the catalytic site in the SWIRM+MPN construct to asparagine (D660N) (Supplementary Figure 7a). Much like SWIRM+MPN, when expressed in *Mysm1*−/− BMDMs, the inactive SWIRM+MPN (D660N) interacted with RIP2 upon L18-MDP stimulation (Supplementary Figure 7b). However SWIRM+MPN (D660N) failed to inhibit the recruitment of TAK1, HOIP, TAB2, TAB3 and, NEMO into the NOD2:RIP2 complex, hence we observed enhanced activation of TAK1, IKKα/β, and MAPK (Supplementary Figure 7b) and induction TNFα and IL6 (Supplementary Figure 7c, d). These results demonstrate that MYSM1 negatively regulates the NOD2 pathway through the cooperative activity of the SWIRM and MPN domains. Both domains are required for interaction with Ub-RIP2 with the latter additionally mediating the cleavage of K63-, K27-, and M1-linked polyubiquitin chains.

**MYSM1 restrains NOD2-mediated inflammation and tissue injury**. Finally, to elucidate the immunological relevance of MYSM1 in NOD2-mediated inflammation, WT and *Mysm1*−/− mice were subjected to a MDP-induced model of peritonitis and systemic inflammation. *Mysm1*−/− mice exhibited higher recruitment of inflammatory cells (neutrophils) not only into the peritoneum (the primary site of MDP challenge), but also into

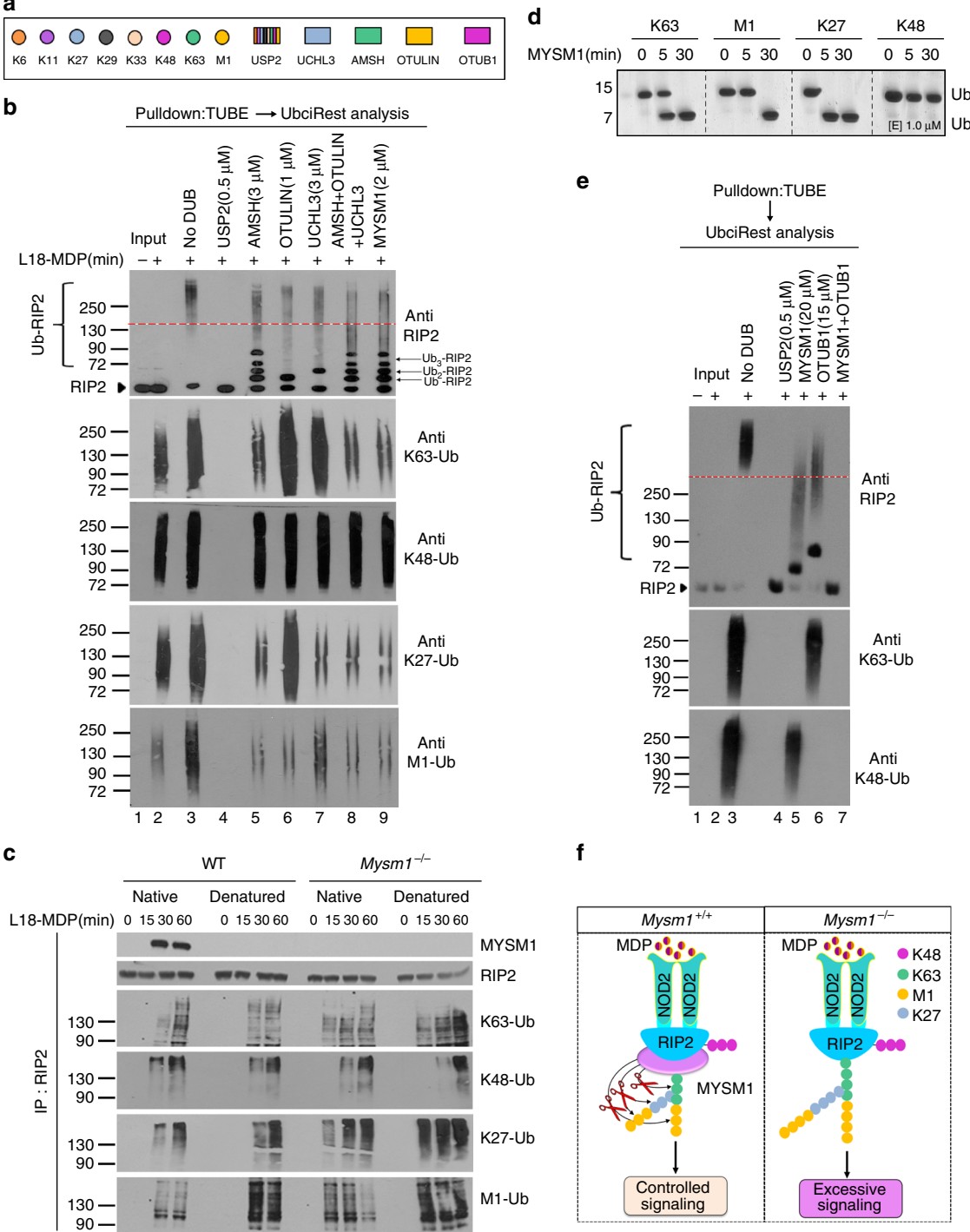

**Fig. 5** MYSM1 disrupts the NOD2:RIP2 complex by removing K63, K27, M1 but not K48 polyubiquitins. **a** Color codes of ubiquitin linkage specificity of DUBs used in the UbiCRest analysis (**b, e**). **b** TUBE pulldowns from BMDMs stimulated with L18-MDP (1 h) were subjected to UbiCRest then immunoblotted for RIP2, K63, K48, K27, and M1 polyubiquitin chains. The red dashed line is an arbitrary line highlighting a relative shift from high to lower MW intermediate /complete digestion products of Ub-RIP2. **c** Native and denatured cytoplasmic fractions from WT and *Mysm1*−/− BMDMs stimulated with L18-MDP for indicated duration were immunoprecipitated with anti-RIP2 and pulldowns were immunoblotted for indicated molecules. **d** Ubiquitin linkage specificity of MYSM1. Di-ubiquitins of indicated linkages incubated with recombinant MYSM1 were separated by SDS-PAGE gel and visualized by silver-staining. **e** TUBE pulldowns from WT BMDMs stimulated with L18-MDP for 1 h were incubated with higher concentration of the indicated DUBs for 3 h. Samples were then immunoblotted for RIP2, K63, and K48 polubiquitin chains. The dotted red lines (**b, e**) are arbitrary lines indicating a relative shift from high to lower MW Ub-RIP2 forms after digestion. Note that for maximum Ub restriction, in **e**, a higher concentration of DUBs was used compared to **b**. **f** Schematic of proposed model for removal of polyubiquitins from RIP2 by MYSM1. Data are representative of two to six independent experiments. See also Supplementary Figure 5

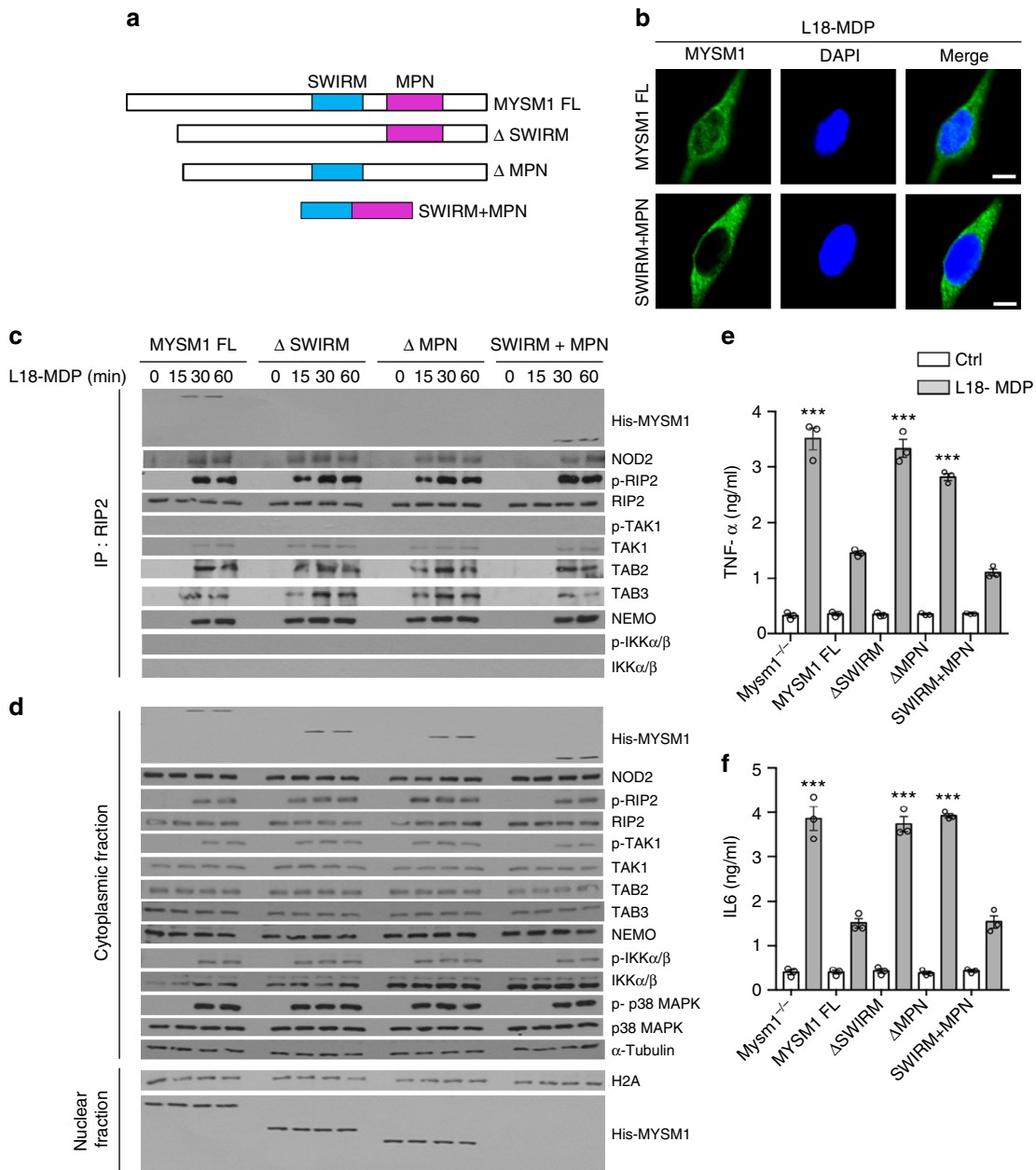

**Fig. 6** MYSM1 inactivates RIP2 complex via the SWIRM and MPN domains. **a** Schematic representation of MYSM1 domain mutant constructs.
**b** Immunofluorescence staining of full-length MYSM1 (MYSM1 FL) or SWIRM+MPN in complemented $Mysm1^{-/-}$ BMDMs 1 h post L18-MDP stimulation.
Scale bar (10 μm). **c**, **d** $Mysm1^{-/-}$ BMDMs complemented with indicated MYSM1 constructs were stimulated with L18-MDP. Cytoplasmic fractions were
immunoprecipitated with anti-RIP2. Pulldowns (**c**) and the corresponding cytoplasmic or nuclear fractions (**d**) were analysed for indicated molecules.
**e–f** $Mysm1^{-/-}$ BMDMs or $Mysm1^{-/-}$ BMDMs complemented with indicated constructs were stimulated with L18-MDP for 12 h and analysed for TNF-α
and IL-6 secretion by ELISA. Results in **e–f** are from three independent experiments. Data are shown as mean ± s.e.m. ($n = 3$). ***$p < 0.001$ determined by
one-way ANOVA followed by Bonferroni's post-test depicts statistical significance relative to Ctrl. See also Supplementary Figure 6 and Supplementary
Figure 7

peripheral organs, including the spleen (Fig. 7a–c). To further assess the impact of MYSM1 in NOD2-mediated inflammation and tissue damage, we resorted to the MDP-induced model of liver injury. For that, WT, $Mysm1^{-/-}$, and $Rip2^{-/-}$ mice were sensitized with galactosamine (GalN), an inhibitor of hepatic protein synthesis, before challenge with MDP. Compared to the WT or the unresponsive $Rip2^{-/-}$ mice, $Mysm1^{-/-}$ mice exhibited enhanced systemic inflammation characterized by higher cytokine (TNFα, IL6) levels, as well as the hepatocellular enzyme alanine aminotransferase (ALT), a marker of liver damage (Fig. 7d, e). To evaluate the specific contribution of macrophages

and monocytes in NOD2-mediated inflammation and tissue injury, we also analyzed the $Mysm1^{fl/fl}$ mice crossed with myeloid-specific Cre deleter LysM-Cre. $Mysm1^{fl/fl}$ LysM-Cre mice also had higher TNFα, IL6, and ALT levels compared to the $Mysm1^{fl/fl}$ controls (Fig. 7f, g). Together, these results revealed MYSM1 as a central negative regulator of NOD2:RIP2 signaling that protects against excessive inflammation and tissue injury.

## Discussion

The NOD2 pathway, a key component of the innate immune system, requires strict regulation to optimize protection against

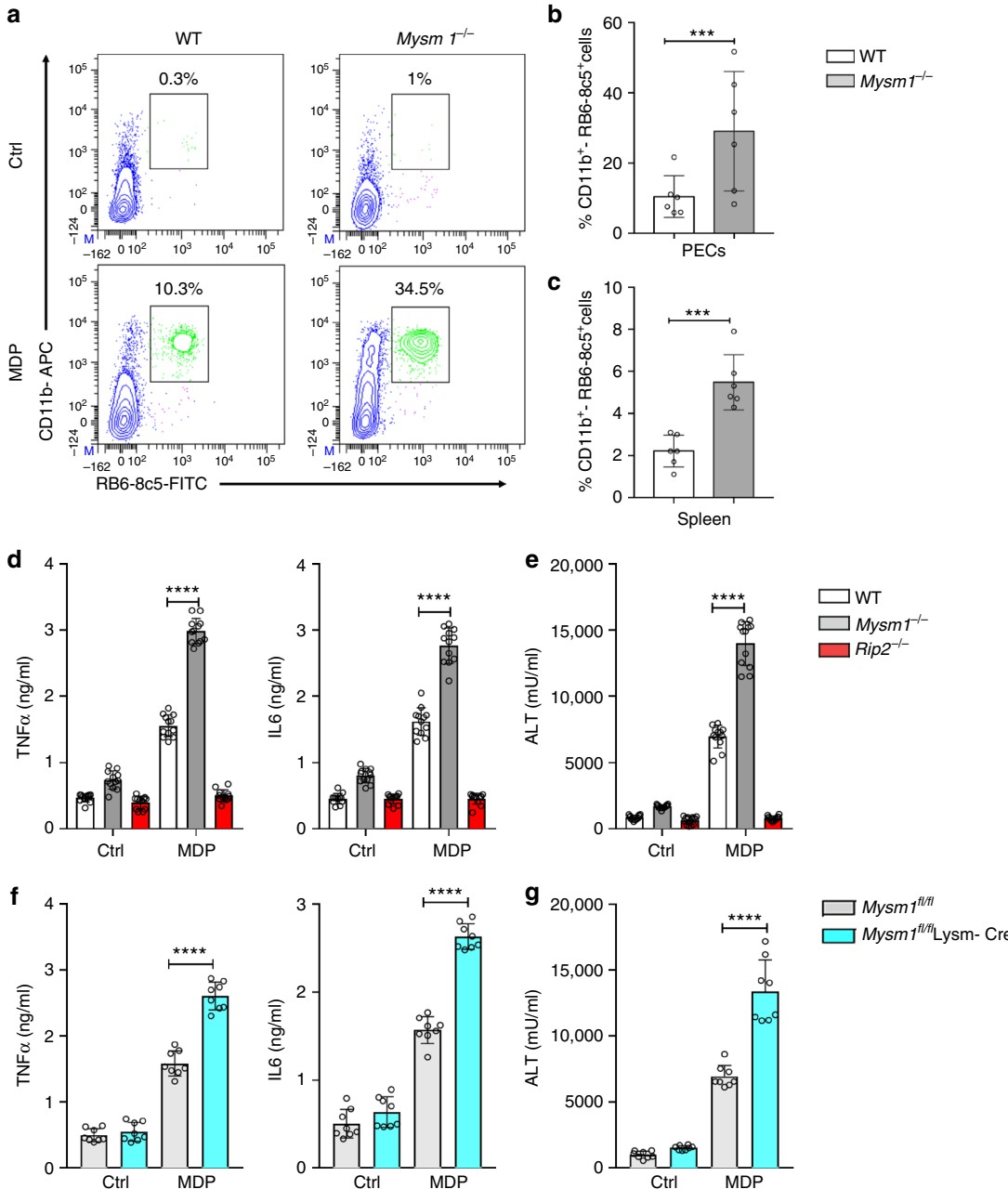

**Fig. 7** MYSM1 suppresses NOD2-mediated systemic inflammation and liver injury. **a** Representative flow cytometry contour plots showing the recruitment of neutrophils (CD11b[+], RB6-8c5[+]) into peritoneal cavities (PECs) of WT ($n = 6$) and $Mysm1^{-/-}$ ($n = 6$) mice 6 h post administration of MDP (25 mg/ kg body weight) or control buffer (Ctrl). **b–c** Corresponding bar graphs showing the mean percentage of neutrophils in PECs (**b**) and spleen (**c**). **d–e** WT ($n = 12$), $Mysm1^{-/-}$ ($n = 12$) and $Rip2^{-/-}$ ($n = 12$) mice were inoculated with GalN (1 g/kg body weight, via intraperitoneal (i.p) route 4 h prior to challenge with MDP (25 mg/ kg body weight, i.p) analysed 6 h later for TNF-α, IL-6 (**d**) and liver alanine aminotransferase (ALT) in the sera (**e**). **f–g** TNF-α, IL-6 (**f**) and ALT (**g**) in sera of $Mysm1^{fl/fl}$ ($n = 8$) or $Mysm1^{fl/fl}$ LysM-Cre ($n = 8$) mice challenged with GalN and MDP as in **d–e**. Each data point represents (**b-g**) an individual mouse. Data are shown as mean ± s.d.. ****$p < 0.0001$ determined by one-way analysis of variance (ANOVA) with Bonferroni's multiple-comparison test. Data in **b**, **c** and **f**, **g** are from two independent experiments, (**d**, **e**), from three independent experiments. See Supplementary Figure 10 for gating strategy for the FACs data in panels (**a–c**)

infections, while preventing a self-destructive inflammation. A focal point of NOD2 signaling is RIP2, a scaffolding protein that nucleates the recruitment and activation of downstream signaling proteins leading to NF-kB- and MAPK-mediated cellular responses. Whereas the ubiquitination of RIP2 is well acknowledged to lie at the heart of NOD2 signaling, the precise nature of the ubiquitin modifications involved and how they are regulated to ensure a coordinated activation and resolution of NOD2-mediated inflammation remains poorly defined. This study

reports two major discoveries. First, we provide a detailed characterization of ubiquitin chain types attached to RIP2. We show that NOD2 activation evokes the attachment of K48-, K63-, and M1-linked polyubiquitin chains, as well as non-canonical K27-linked polyubiquitin chains, onto RIP2. We reveal the sequence by which these different chains are attached to RIP2 and their structure. Secondly, we identify MYSM1 as the proximal deubiquitinase that attenuates NOD2:RIP2 signaling complex assembly. We show that although exclusively in the nucleus at steady

state, in response to NOD2 activation, MYSM1 is rapidly recruited to the NOD2:RIP2 complex, wherein it acts as a Swiss army knife selectively cleaving the pro-signaling K63-, K27-, and M1-linked polyubiquitin chains, but not the degradative K48-linked chains. Finally, we provide the in vivo physiological significance of this regulation by demonstrating that whole body or myeloid–specific ablation of MYSM1 in mice causes unrestrained NOD2-mediated inflammation and tissue injury.

These findings fill an important gap in our understanding of how the NOD2 pathway is activated and regulated. While CYLD[25–28], OTULIN[28–30,40,42], A20[25,33] have previously been reported to regulate this pathway, all of them require LUBAC to impede on NOD2 signaling[25–28,31,33]. Thus, how the initial polyubiquitination events driving NOD2:RIP2 complex assembly are controlled has been unclear. The results herein show that upon NOD2 triggering, independently of LUBAC activity, MYSM1 rapidly accumulates in the cytoplasm, interacts with and removes K63-, K27-, and M1-linked polyubiquitins from RIP2 (Fig. 5f). This disrupts the recruitment of proximal signaling molecules including XIAP, LUBAC, NEMO, and TAK1 into NOD2:RIP2 complex and thereby attenuates downstream NF-kB and MAPK-mediated inflammatory responses (Fig. 3). Remarkably, we found no direct interaction between MYSM1 and LUBAC or TAK1 or the IKK complex (NEMO, IKKα/β) (Figure 2c. 1 and Fig. 4b). Moreover, in vitro, Rec-MYSM1 could remove polyubiquitins from RIP2, but not NEMO (Supplementary Figure 6b, c). Together, these results establish MYSM1 as a central negative regulator of the NOD2 pathway acting upstream of the previously described major regulators CYLD, OTULIN, and A20. Importantly, they demonstrate that MYSM1-mediated negative regulation is due to its direct effect on RIP2 and not on the downstream effectors, such as LUBAC or NEMO.

How does MYSM1 deubiquitinate RIP2? The results herein show that K63-, K27-, and M1-linked polyubiquitin chains on RIP2 are part of a hybrid polymers with the K63 linkages being the initial chains onto which K27 and M1-chains are subsequently attached in that sequence (Fig. 1 and Fig. 5). Based on our findings, we surmise that restriction of these hybrid chains by MYSM1 most likely occurs both by cleaving the proximal K63 chains, as well as by trimming the appended K27 and M1 chains. Regardless, the present data demonstrate MYSM1 as a DUB uniquely positioned for effective disruption of the upstream NOD2 signaling events; it directly interacts with RIP2 and specifically removes the non-degradative K63-, K27-, and M1-linked polyubiquitins essential for signal propagation, but spares the K48-polyubiquitins that target the NOD2:RIP2 complex for degradation[43,44].

Whereas ubiquitin modifications are accepted to regulate virtually all biological processes, the ubiquitin research has largely focused on the K48- and K63-linked chains[6]. More recently, a better understanding of the M1-linked chains, particularly in immune signaling is beginning to emerge[12,13,45]. In spite of the few reports linking K27-linked chains to DNA damage[46] and immune signaling pathways[47–49] or the K6-linked chains to DNA damage[50] and mitophagy[51], generally, the function of these and other atypical ubiquitin chains and the enzymes (E3 ligases and DUBs) that conjugate or remove them remain largely elusive. For instance, apart from OTULIN[32,52] and CYLD[25,27], very little is known about other DUBs that can hydrolyze M1 linkages. We now show that in addition to K63 chains, MYSM1 also efficiently cleaves the non-canonical M1, K6, and K27 chains relevant for its possible involvement in other biological processes. In sum, this study demonstrates MYSM1 as a critical negative regulator of innate immunity and its mechanism. Finally the complete description of the ubiquitin linkage specificity of MYSM1

described herein highlights the possible utility of this DUB as a cell biology tool in the ubiquitin research.

## Methods

**Ethics statement.** All mice were maintained under specific pathogen-free conditions and experiments were carried out according to the guidelines set out by the Jordbruks Verket (approval no. A53-14).

**Animal experiments.** Animal experiments were performed in Umeå Centre for Comparative Biology (UCCB). Mice were inoculated intraperitoneally (i.p) with MDP (25 mg/kg body weight) and 6 h later analysed for recruitment of inflammatory cells into the peritoneum and spleen. For the liver injury model, mice were first sensitized with GalN (1 g/Kg body weight) via i.p route 4 h prior to i.p injection of MDP (25 mg/kg body weight). 6 h later they were analysed for cytokine or ALT levels in serum. Experiments were performed using adult mice (8–12-week-old).

**Mice.** For germline deletion of *Mysm1*, male *Mysm1* floxed mice (*Mysm1*-A04; *Mysm1*™1a(ΔMP)Wtsi MGI#: 2444584 from KOMP Repository at UC Davis) were bred with female transgenic K14- Cre mice[53]. K14-Cre^{wt/tg}*Mysm1*^{flox/flox} were further crossed with each other to generate *Mysm1*^{−/−} mice without Cre allele[36]. For myeloid-specific ablation of Mysm1, *Mysm1*^{flox/flox} were bred with LysM-Cre transgenic mice to generate *Mysm1*^{flox/flox} LysM-Cre[36]. Breeding of mice was done at the Umeå Transgene Facility (UTCF), whereas in vivo infection experiments were performed in Umeå Center for Comparative Biology (UCCB).

**Reagents and antibodies.** Anti-p-RIP2 (4364, 1:1000),anti-RIP2 (4142, 1:1000), anti-p-TAK1(9339, 1:1000), anti-TAK1 (4505, 1:1000), anti-p-IKKα/β (2078, 1:1000), anti-p-p38MAPK (4631, 1:1000), anti-p38 MAPK (9212, 1:1000), anti-xIAP (2042, 1:1000), anti-H2A (12349, 1:1000), anti-K63 (5621, 1:1000) and anti-K48 (8081, 1:1000) were purchased from Cell signaling technology. Anti-MYSM1 (ab193081, 1:3000), anti-HOIP (ab85294, 1:500), anti-IKKα/β (ab178870, 1:1000) and anti-K27 (ab181537, 1:2000) were purchased from Abcam. Anti-TAB2 (MA514818,1:500), RBC-8c5 (35-5931-80, 1:500) NuPAGE™ Novex™ 4–12% Bis Tris Midi protein gels (WG1402), NuPAGE MES SDS running buffer (NP0002) and ECL Western Blotting Detection Kit were purchased from Thermo-Scientific. Anti-NEMO (559675,1:1000) and CD11b (553312,1:1500) were purchased from BD Biosciences. UbiCREST diubiquitinase enzyme set (K-400), Agarose TUBE 2 (UM402), anti- K63 TUBE; FLAG (UM604, 500 nM), anti- M1 (linear) TUBE; FLAG (UM606, 500 nM), anti-M1 (AB-130,1:1000) were purchased from Lifesensors. Anti-FLAG® M2 affinity gel (A2220), anti α- tubulin (T3828, 1:5000) ALT activity assay kit (MAK052), LPS (L4391) and Gilotoxin (G9893) were purchased from Sigma-Aldrich. Anti-TAB3 (sc-46550, 1:500), anti-NOD2 (sc-30199, 1:500), protein A/G plus agarose bead (sc-2003), HRP-conjugated goat anti-mouse IgG (sc-2005, 1:5000), goat anti-rabbit IgG (sc-2004, 1:5000) and donkey anti-goat IgG (sc-2020, 1:5000) were purchased from Santa Cruz biotechnology. L18-MDP (tlrl-imdp) and MDP (tlrl-mdp) were from Invivogens. X-ray films were purchased from Kodak.

**Cell culture and stimulation.** BMDMs were generated by culturing bone marrow progenitors in IMDM (1×) +Glutamax™-I (Gibco, Life technology) supplemented with 10% FCS (Gibco, Life technology), 100 U/ml penicillin/streptomycin (Sigma-Aldrich) and 20% (v/v) L929 conditional medium for 5 days. For inflammatory response, BMDMs were stimulated by adding either LPS (E. coli 0111: B4, Sigma-Aldrich, 500 ng/ml) or L18- MDP (Invivogen, 200 ng/ml) directly to the culture medium. For inhibitor studies, BMDMs were pretreated with (or without) the LUBAC inhibitor GT (500 ng/ml) or the SMAC mimetic (cIAP1/2 inhibitor, LCL-161) (100 nM) for 18 h prior to L18-MDP stimulation.

**Immunoprecipitation and immunoblot analysis.** Cells were lysed in 2× Laemmli buffer then analysed directly by SDS-PAGE immunoblotting. For immunoprecipitation, cells were suspended in lysis buffer (1% NP-40, 50 mM Tris-HCl [pH 7.5], 150 mM NaCl, 1 mM NaF, 2 mM PMSF, protease inhibitor cocktail [Roche Applied Science], 1 mM sodium orthovanadate and 10 mM sodium pyrophosphate). For denaturation, lysates were suspended in 100 µl TSD buffer (50 mM, Tris pH 7.5, 1% SDS, and 5 mM DTT) and then boiled for 10 min. After centrifugation (13,000 rpm for 5 min at room temperature), to clear the lysates, samples were diluted in 1.2 ml of TNN buffer (50 mM Tris [pH 7.5], 250 mM Nacl, 5 mM EDTA, and 0.5% NP40). Samples were incubated with specific antibodies at 4° overnight with rotation and the complexes were pulled down using Protein A/G agarose bead. Agarose beads were then washed three times with protein lysis buffer containing the standard salt concentrations, suspended in Laemmli buffer and analysed by SDS-PAGE immunoblotting. For ubiquitination assays, cells were lysed with ubiquitination lysis buffer (8 M urea, 50 mM Tris [pH 7.5], 25 mM NaCl, protease cocktail inhibitors, 5 mM EDTA, 2 mM N-ethylamide). After sonication, lysates were centrifuged (14,000 rpm for 10 min) to remove precipitates. Samples were diluted in reducing SDS sample buffer then boiled at 70 °C for 10 min before immunoblotting with anti-ubiquitin antibodies.

**Flow cytometry**. For staining, single- cell preparations were made from spleen and PECs in FACs buffer (PBS with 2% FBS). Staining panel containing CD11b and RBC-8c5 were performed in FACs buffer. After ACK (ammonium chloride) lysis of RBC, cells were washed once and were incubated with antibodies for 45–60 min on ice. Flow cytometry was performed on a LSRII (BD Biosciences) and data were analysed using the BD FACSDIVA software. Analysis was done according to the gating strategy depicted in Supplementary Figure 10.

**Subcellular fractionation**. $5 \times 10^6$ BMDMs were washed, scraped with cold DPBS (Dulbecco's PBS), pelleted and resuspended in cold buffer A (250 mM sucrose, 250 mM KCl, 5 mM MgCl$_2$ and 50 mM Tris-HCl [pH 7.4]), lysed by gentle pipetting while adding NP-40 to a final concentration of 0.7%, and further incubated on ice for 10 min. The suspension was centrifuged at $750 \times g$ for 10 min at 4 °C and the resulting pellet containing nuclei was lysed in NP40 lysis buffer for biochemical analysis.

**Immunofluorescence microscopy**. BMDMs cultured on coverslips were fixed with 4% (w/v) paraformaldehyde in PBS for 15 min, permeabilized with 0.2% (v/v) Triton X-100, blocked with 3% (w/v) BSA in PBS for 30 min, incubated with anti-MYSM1 antibody containing 1% (w/v) BSA and 0.1% (w/v) saponin in PBS for overnight at 4 °C, washed with PBS, and then incubated with anti-rabbit IgG Alexa 488 fluorescence secondary antibody for 1 h. Following washing, cells were stained with DAPI, washed with double distilled water, mounted onto microscope slides and imaged using a NIKON C1 confocal microscope.

**Purification of endogenous Ub conjugates**. Ub conjugates were purified using affinity reagents TUBE, K63-TUBE and M1-TUBE (Lifesensors). Briefly, cells were lysed in TUBE lysis buffer (50 mM Tris [pH 7.5], 0.15 M Nacl,1 mM EDTA, 1% NP-40, 10% Glycerol and supplemented with 5mM N-ethylmaleimide, 20 μM PR619 [pan DUB inhibitor], 1× protease cocktail inhibitor (Roche), lysates were cleared by centrifugation. For TUBE pulldowns, agarose-TUBEs were equilibrated according to manufacturer's instructions (Lifesensors), then added directly to the cell lysates and incubated agitating at 4 °C overnight. Agarose-TUBE beads were then washed three times in ice-cold TUBE lysis buffer and suspended in Laemmli buffer. For K63- and M1-TUBE pulldowns, lysates were incubated with 500 nM of anti-K63 TUBE FLAG and anti-M1 TUBE FLAG at 4 °C overnight with rotation and the complex were then pulled down by anti-FLAG affinity gel (Sigma-Aldrich). The beads were washed three times in ice-cold TUBE lysis buffer and the bound material was eluted with 1x Laemmli buffer.

**Expression and purification of recombinant MYSM1**. Mouse MYSM1 gene with C-terminal His tag (His-MYSM1) or His-SWIRM-MPN fusion were cloned into the pET30a expression plasmid and subsequently transformed into *E. coli* BL21 (DE3). Bacteria were cultured in 1 L M9 medium and induced with 1 mM IPTG at 15 C overnight. Cell pellets were lysed by sonication in the buffer 50 mMTris-HCl, pH 8.0 with PMSF. The supernatant after centrifugation was purified by Nickel column. His-MYSM1 was eluted with a stepwise gradient of imidazole.

**Deubiquitinase assays**. Ub conjugates from unstimulated or L18-MDP treated BMDMs were isolated by TUBE as described under purification of endogenous ub conjugates. After washing, the beads were re-suspended in 1× DUB dilution buffer. For Ubiquitin restriction analysis, Ub conjugates incubated without or with DUBs (USP2 [0.5 μM], OTUD3 [3 μM], Cezanne [2 μM], UCHL3 [3 μM], Trabid [3 μM], OTUB1 [10 μM], AMSH [3 μM], OTULIN [1 μM] and MYSM1 [2 μM]). All samples were incubated at 37 °C for 1 h with shaking and reaction was stopped by adding 2× Laemmli buffer. Samples were then resolved in 4–12% Nupage gradient gel (Life technology), followed by western blot analysis with anti- RIP2, and specific antibodieas against Ubiquitin linkages (M1, K27, K48, and K63). Representative uncropped ubiquitin blots for Fig. 1 and Fig. 5 are provided in Supplementary Figure 8 and Supplementary Figure 9 respectively.

For in vitro deubiquitinase assay, beads were resuspended in 1× Deubiquitinase buffer (125 mM HEPES ([pH 7.5], 25 mM MgCl2, 10 mM NaF, 10 nM okadaic acid, 500 mM NaCl, 2.5 mM DTT, 0.5 mg/ml BSA). 10 μl of recombinant His-MYSM1 (1 μg) was then added to the bead slurry, followed by incubation for 30 min at 37 °C with gentle shaking. The reaction was then stopped by adding 2x SDS loading buffer followed by western blot analysis for anti K63, K48, K27, and M1 ubiquitin antibody.

In vitro ubiquitin linkage specificity assay was performed as previously described[54]. Briefly, DUB assay buffer (150 mM NaCl, 25 mM Tris [pH 7.5], and 10 mM DTT) was activated at 23 °C for 10 min. 2× concentrated DUB stock was prepared by diluting MYSM1 in DUB assay buffer and activated at 23 °C for 10 min. Subsequently, 5 μl of diluted enzyme was mixed with 1–2 μg of K6, K11, K27, K29, K33, K48, K63, and M1 diubiquitins along with 2 μl of 10× DUB buffer in 20 μl of reaction mixture. At different time points (5 and 30 min) the enzymatic reactions were stopped by addition of 4 μl SDS sample buffer containing 100 mM DTT. 5 μl aliquots were then resolved on 4–12% Nupage gradient gel run in MES buffer (Life Technology). Ubiquitin cleavage was detected by silver staining using the Sliver Stain Plus kit (Bio-Rad).

**Plasmids and retroviral-mediated ectopic gene expression**. Full-length (FL) MYSM1, deletion mutants of MYSM1 or fusion construct (SWIRM+MPN) with a C-terminal His tag were cloned into the retroviral ectopic expression vector pBabe puro IRES EGFP (Addgene plasmid #14430). Similarly, for expression of RIP2 variants, FLAG-tagged full-length RIP2 (RIP2) or arginine/lysine substituted RIP2 (RIP2 [K209R]) were cloned into the pBabe puro IRES EGFP.

For ectopic gene expression, the above constructs were co-transfected with the pCL-Eco (Addgene plasmid #12371) into HEK293T cells (ATCC CRL-3216™). Cell supernatants were harvested after 36, 48, and 72 h after transfection, pooled and virus particles concentrated by ultracentrifugation for 4 h in a SW28 rotor at 24,000 rpm ($76,000 \times g$). Virus particles were resuspended in 200 μl of Optimem then either stored at −80 °C or used to infect cells. For ectopic expression in BMDMs, 10 μl of concentrated retroviral particles were added onto bone marrow progenitors ($10^6$ cells/well in a 6 well plate), in the presence of polybrene (4 μg/ml), centrifuged for 1 h at 2000 rpm (700 x $g$) at RT, washed with PBS and then cultured with complete IMDM medium with 20% L929 conditioned medium for 6 days.

**ELISA and quantitative real time PCR analysis**. TNF-α and IL-6 were assayed using mouse R&D Systems ELISA kit. RNA was isolated from cells using RNeasy kit from Qiagen. 1 μg of total RNA was reverse transcribed using oligo(dT) primers (New England Biolabs). Quantitative Real-time PCR analysis was performed and analyzed using ABI Prism 7500 Fast RT-PCR System (Applied Biosystems). The results were normalized to the housekeeping genes and expressed as fold change relative to RNA samples from control or mock-treated cells/mice using the comparative CT method (ΔΔCT). All Taqman gene expression data are presented as the expression relative to 18 S rRNA reference gene (Rn18s). The following Taq-Man Gene Expression probes were used (Applied Biosystems): *Tnf* (Mm00443258_m1), *Il6* (Mm00446190_m1) and *Rn18s* (Mm03928990_g1).

**Statistical analysis**. Data in text and figures are expressed either as mean with standard error of the mean (±s.e.m.) or mean with standard deviation (±s.d.). Statistical comparison between WT and *Mysm1*$^{-/-}$ cells or mice treated with NOD1/NOD2 ligands was done using one-way ANOVA with Bonferroni post-test. *$P$-values < 0.05, **$P$ < 0.01, ***$P$ < 0.001; were considered to be significant.

## Data availability

The data supporting the findings of this study are available within the paper and its supplementary information files and can be provided by the authors upon reasonable request. A reporting summary for this Article is available as a Supplementary Information file.

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

## Acknowledgements

This work was funded by the Laboratory for Molecular Infection Medicine Sweden (MIMS), the Medical Faculty, Umeå University, the Swedish Research Council (VR grants 2015-02857 and 2016-00890 to N.O.G), the Swedish Cancer Foundation (grant, CAN 2017/421 to N.O.G). We thank Barbara Sixt, Umeå University for critical comments on the manuscript.

## Author contributions

S.P. designed and performed all experiments. N.O.G. conceived and supervised the study, performed in vivo experiment and flow cytometry analysis. S.P and N.O.G prepared and approved the manuscript.

## Additional information

**Competing interests:** The authors declare no competing interests.

