## [Peer Review file · Nature Communications]

(Reviewer #1)

This is the revised version of their manuscript and is substantially rewritten (new order of figures, which I think is better than the previous version). It also addresses criticisms related to the UbiCREST assays. The explanations in the text now help understand the interpretations better. I recommend that in the Discussion the authors state the limitations of the UbiCREST assay (enzyme specificities, lack of enzymes that only cleave K33 or K29 and reaction conditions) and that future experiments with mass spectrometry should be carried out for validation. My major concern in this version is that quantitative data have been incorrectly analyzed, which should be corrected as below. I have seen this error before and I hope, unlike in other cases, these authors do not cite other manuscripts where analyses have been wrongly performed to justify it. Based on the way the data appear on graphs, I am quite sure that overall interpretations are unlikely to change i.e. Mysm1^{-/-} cells have means on average higher than WT cells. These should be corrected before publication.

Major comments

1. The authors were asked to explain technical vs biological replicates in Figure Legends, and to clarify number of mice used in in vivo experiments. While the explanation related to mouse experiments are satisfactory, they seem to have misunderstood "All data points should be shown wherever possible in other figures also.". The authors have shown 9 data points from 3 tech replicates from 3 experiments, which is incorrect because statistics would have used 9 independent & non-independent means and generated lower P values than what would have been obtained with 3 independent means. The authors have made the error of showing/using all technical replicates from in vitro experiments in the final analyses when they should have only shown/used means of independent experiments – the way it stands now, there appears to be pseudoreplication. They should generate one mean per experiment derived from technical replicates but, technical replicates should not be used for subsequent statistical analyses across experiments. Done this way (which is the only correct way because technical replicates from in vitro experiments are not biologically independent), n=3 biologically independent experiments will generate only 3 datapoints which should be used for graphing and analysis. For a good discussion on this see EMBO Rep <https://www.ncbi.nlm.nih.gov/pmc/articles/PMC3321166/>
2. It is also unclear to me why the authors use one-way ANOVA when they clearly have two factors. For example, in Fig 3A-D – there are three genotypes and two treatments (2 factorial ANOVA), but they use one-way analyses. Did they only analyse L18-MDP or C12-iE-DAP treated samples? If so, they should state this in the analyses and the reasoning behind this test and not including untreated condition.
3. Same applies to the analyses of Fig 7D-G – there are 2 genotypes and 2 conditions for each, yet one-way ANOVA are used here.

(Reviewer #2)

Although the authors referred Fig.1 in response to my comment, I believe that they are referring Fig. 7 instead of Fig. 1. The authors have addressed my comment by the findings showing in Fig. 7, and so the paper is publishable.

Point by point responses to Reviewers' comments

Reviewer #1 (Remarks to the Author):

This is the revised version of their manuscript and is substantially rewritten (new order of figures, which I think is better than the previous version). It also addresses criticisms related to the UbiCREST assays. The explanations in the text now help understand the interpretations better. I recommend that in the Discussion the authors state the limitations of the UbiCREST assay (enzyme specificities, lack of enzymes that only cleave K33 or K29 and reaction conditions) and that future experiments with mass spectrometry should be carried out for validation. My major concern in this version is that quantitative data have been incorrectly analyzed, which should be corrected as below. I have seen this error before and I hope, unlike in other cases, these authors do not cite other manuscripts where analyses have been wrongly performed to justify it. Based on the way the data appear on graphs, I am quite sure that overall interpretations are unlikely to change i.e. Mym1^{-/-} cells have means on average higher than WT cells. These should be corrected before publication.

Response: We thank the reviewer for the constructive comments. We have included the suggestions. Please see page 5, lines 16-20.

Major comments

1) The authors were asked to explain technical vs biological replicates in Figure Legends, and to clarify number of mice used in in vivo experiments. While the explanation related to mouse experiments are satisfactory, they seem to have misunderstood “All data points should be shown wherever possible in other figures also.”. The authors have shown 9 data points from 3 tech replicates from 3 experiments, which is incorrect because statistics would have used 9 independent & non-independent means and generated lower P values than what would have been obtained with 3 independent means. The authors have made the error of showing/using all technical replicates from in vitro experiments in the final analyses when they should have only shown/used means of independent experiments – the way it stands now, there appears to be pseudoreplication. They should generate one mean per experiment derived from technical replicates but, technical replicates should not be used for subsequent statistical analyses across experiments. Done this way (which is the only correct way because technical replicates from in vitro experiments are not biologically independent), n=3 biologically independent experiments will generate only 3 datapoints which should be used for graphing and analysis. For a good discussion on this see EMBO Rep <https://www.ncbi.nlm.nih.gov/pmc/articles/PMC3321166/>.

Response: We have replotted the data as suggested. Please see modified graphs in Fig. 3a, b, Fig. 6e, f. and Supplementary Figures S1a, S2a,b, S6b,c and S7c,d. Corresponding text changes in the legends are highlighted in yellow.

2) It is also unclear to me why the authors use one-way ANOVA when they clearly have two factors. For example, in Fig 3A-D – there are three genotypes and two treatments (2 factorial ANOVA), but they use one-way analyses. Did they only analyse L18-MDP or C12-iE-DAP treated samples? If so, they should state this in the analyses and the reasoning behind this test and not including untreated condition.

3. Same applies to the analyses of Fig 7D-G – there are 2 genotypes and 2 conditions for each, yet one-way ANOVA are used here.

Response: The objective of the experiments was to determine the difference between WT and *Mysm1*^{-/-} following NOD1/NOD2 stimulation. This is why we only included the statistical comparison for WT vs *Mysm1*^{-/-} cells/mice treated with MDP or C12-iE-DAP. The statistic comparisons for untreated conditions or the *Rip2*^{-/-} samples was considered not necessary because, clearly there was no response in these negative controls. Therefore, the one-way ANOVA is the appropriate method for our comparison. We have now clearly stated this in the Methods section. Please see page 29 lines 13-14 (highlighted in yellow).

Reviewer #2 (Remarks to the Author):

Although the authors referred Fig.1 in response to my comment, I believe that they are referring Fig. 7 instead of Fig. 1. The authors have addressed my comment by the findings showing in Fig. 7, and so the paper is publishable.

Response: We thank the reviewer for his/her contribution towards the improvement of this manuscript.

REVIEWERS' COMMENTS:

Reviewer #1 (Remarks to the Author):

Comments have been adequately addressed.

POINT BY POINT RESPONSE TO REVIEWERS

Reviewer #1 (Remarks to the Author):

Comment: Comments have been adequately addressed.

Response: We are grateful to this reviewer's input on our manuscript and for supporting its publication